# Making, Using, Disposing, Remaking...: Sacred Arts of Re-Creation in Southern Asia

**Susan S. Bean**

Independent Researcher, Cambridge, MA 02139, USA; susan.s.bean@gmail.com

**Abstract:** For centuries, in the eastern Indian subcontinent, areas now in Bangladesh and the Indian states of West Bengal, Odisha, and Bihar, temporary polychrome terracruda (air-dried clay) figural images have been created for periodic *pujas* (rituals of worship) and immersed in nearby rivers or ponds at the event's close. This essay explores how the perennial re-creation of terracruda ritual images supported the rise of goddess worship, stimulated the expansion of the annual cycle of religious festivals, and contributed to a modernizing cosmopolitan public culture. Drawing on recent reconsiderations of materiality that recognize the active roles of inanimate objects and substances, terracruda sacred sculpture is approached through the medium to consider the distinctive contributions that clay makes in interactions with artists, patrons, devotees, and the public. This essay focuses on how the transformational character of air-dried clay enables practices of making, worshipping, and disposing that evoke cosmic cycles, harness potencies that inhere in earth, and realign religious practices in changing times.

**Keywords:** temporary ritual images; Bengal; eastern India; clay; terracruda; materiality; ephemerality; goddess worship; colonial; modernity





## 1. Introduction

In the Indian states of West Bengal, Odisha, and Bihar, and adjacent areas of Bangladesh, temporary ritual images made of terracruda (air-dried clay) have long been in use. Among sculptural mediums, terracruda stands out for its radically transformational character—its penchant to move between states of hardness, plasticity, and fluidity. Because air-dried clay forms are intrinsically ephemeral, physical traces of older images are rare, but early literature indicates a long and widespread presence of terracruda sacred images installed in shrines and temples or made for temporary use.

In Bengal, around 1600 CE, several leading *zamindars* (landlords of extensive estates) served as rulers of their territories in exchange for remitting a share of the revenues to the Mughal governor of Bengal. Some of these powerful zamindars who were devotees of the goddess *(Shaktas)* began to sponsor a new autumn festival for the goddess's manifestation as Durga. Over several days, in the presence of the ruling family and many invitees, priests conducted elaborate rituals involving the recitation of sacred verses and prayers, and offerings of many kinds. These Shakta zamindars elevated the importance of the goddess in her many forms and promoted her worship among their subjects, propagating new manifestations, adding to the annual round of periodic pujas, and popularizing the worship of ephemeral polychrome terracruda images (Figure 1).

In the late 1700s, decades of political and social turmoil culminated in the British East India Company taking full control of the region and systematically eclipsing the zamindari rule. By this time, goddess worship was well established in an annual round of religious festivals. With the shift in political fortunes, new elites and townsfolk prospering under British rule became leading proponents of periodic festivals, staging pujas in their palatial residences and sponsoring community-supported *(barowari)* pujas in their neighborhoods. These occasions for religious devotion also served as opportunities to boost reputations with alluring ritual images, attractive displays, and entertaining performances.

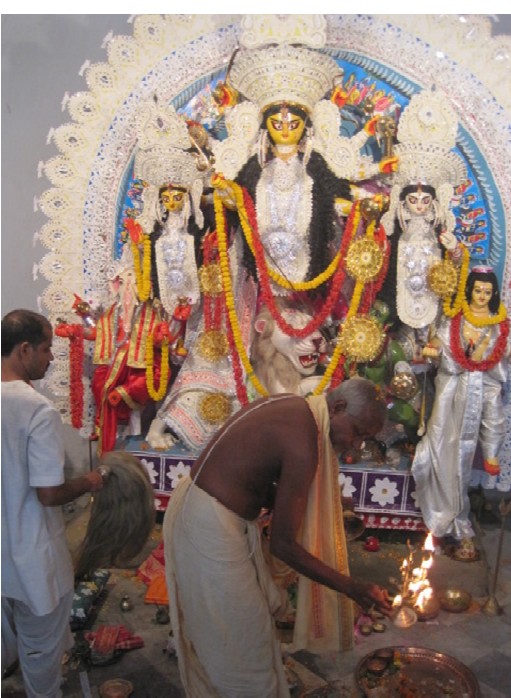

**Figure 1.** Priests performing rituals in front of a traditional terracruda Durga Puja image made for the Akrur Dutt family in the Kumartuli workshop of China Pal, 2012. Author's photograph.

As the nationalist movement advanced in the early 1900s, the festivals gained new prominence as public events for all the people *(sarbojanin)* and the festival cycle increased in popularity after independence in 1947, a trajectory of growth that has continued into the twenty-first century. In the 1990s, the Indian government's removal of many restrictions on private enterprise and international trade prompted some ambitious puja associations to organize conceptually coordinated, skillfully executed displays dubbed 'theme pujas,' with financial backing from commercial sponsors and implementation by teams of art and design specialists (Figure 2).

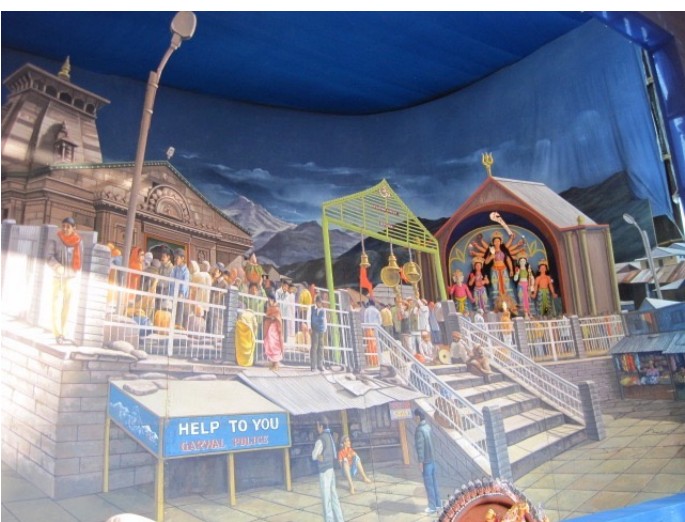

**Figure 2.** Terracruda image of Durga, at right, placed in front of a life-size painted scene depicting a small Himalayan pilgrimage town. Manoharpukur Sarbojanin Puja, Kolkata 2012. Author's photograph.

From the rise of goddess worship in the 1600s to the present, polychrome terracruda figures have been central to the growth of these periodic pujas, sometimes accompanied by other terracruda figures depicting legendary heroes, proverbs, or public notables (Figure 3).

The practice of immersing ritual images or leaving them to decompose once rituals were complete ensured the need for new images and opportunities for adaptation and response to changed circumstances (Figures 4 and 5). This essay considers terracruda's contribution as an exceptionally nimble and ephemeral sculptural medium actively facilitating the creation of new representations of deities and contributing to the expansion of periodic festivals. The distinctive capabilities of terracruda aided clay sculptors as they devised technical, iconographic, and stylistic changes in response to seismic events that were altering power structures and economic systems, and reordering social life from the period of Mughal ascendancy through British rule, postcolonial realignments, and late twentieth-century economic liberalization.

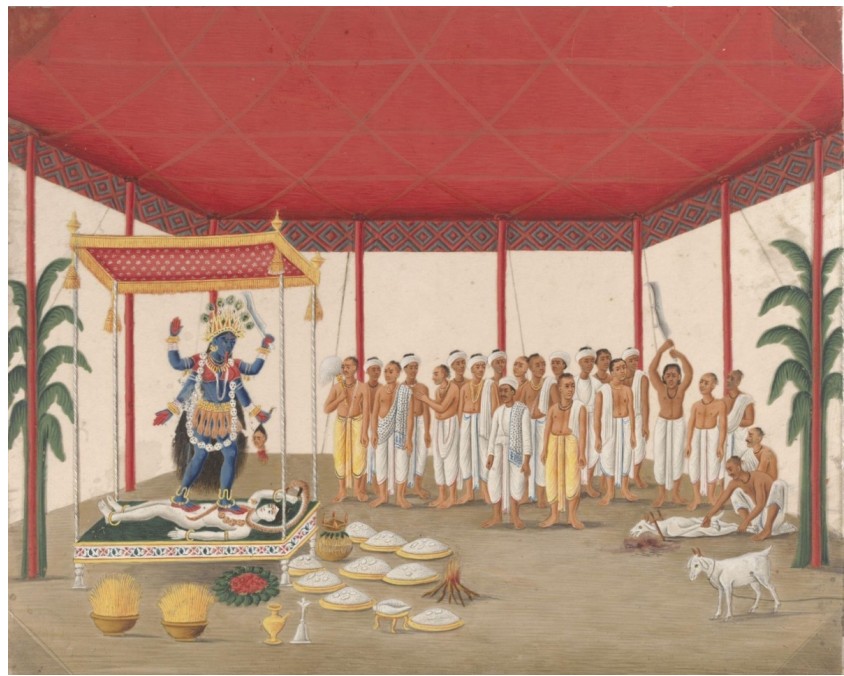

**Figure 3.** Kali Puja being performed in a *pandal* (temporary pavilion) to a terracruda image of the goddess. Gouache on mica 18.5 × 23 cm. Patna, ca. 1860. Courtesy Victoria & Albert Museum.

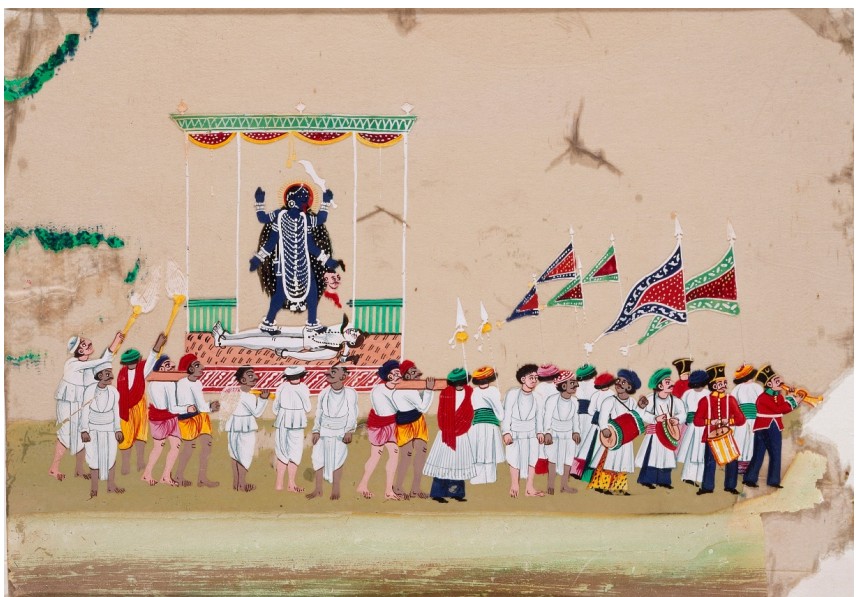

**Figure 4.** Procession escorting a terracruda image of Kali to immersion. Opaque watercolor on mica, 15.24 × 19.69 cm. Murshidabad, c. 1850. Collection Los Angeles County Museum of Art. Image, public domain.

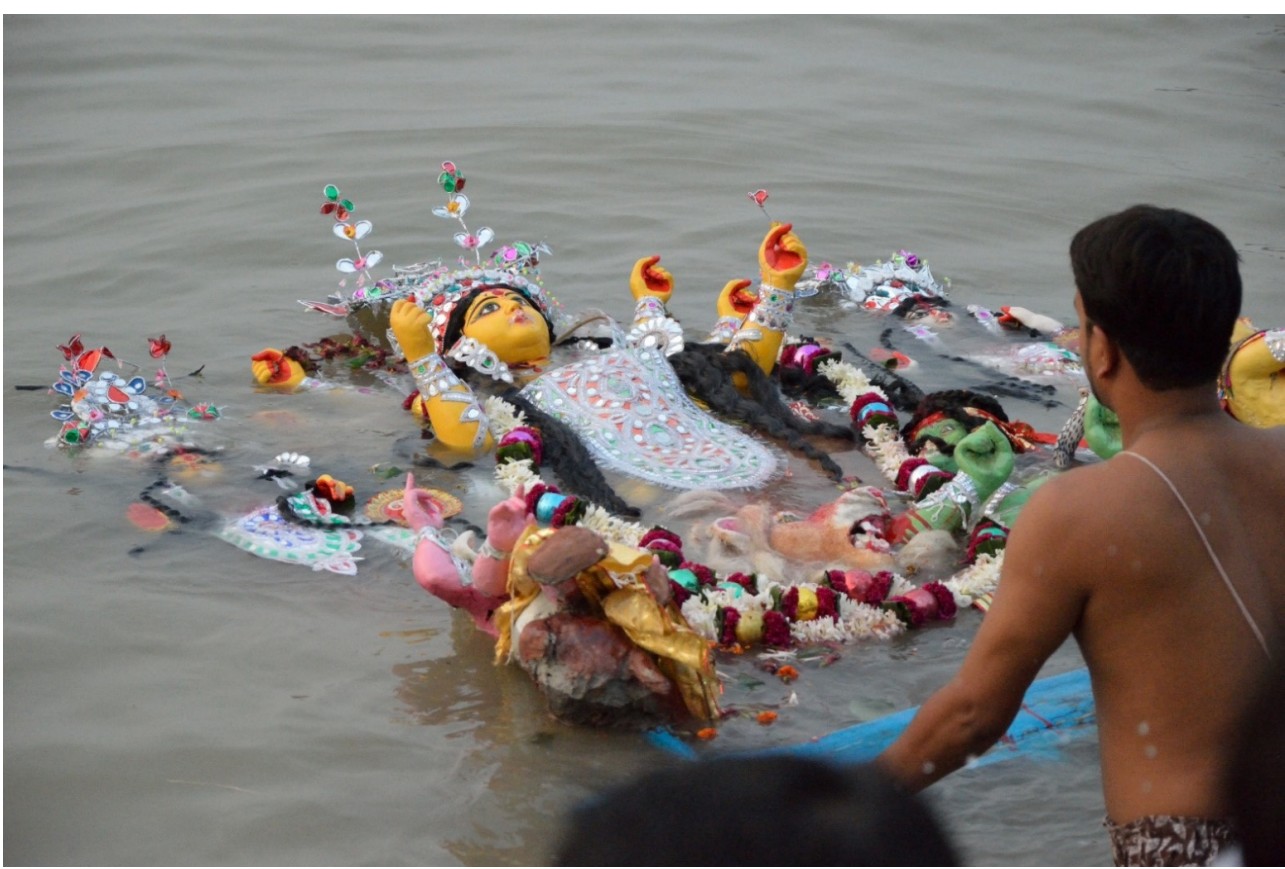

**Figure 5.** Durga Immersion, 3 October 2013, Bagbazar Ghat Kolkata. Photographer Biswarup Ganguly, Wikimedia Commons.

## 2. Background and Methodology

This essay is part of a larger project on terracruda sculpture in southern Asia where practices have deep historical roots and continue to the present. At the heart of these practices are figural representations of divine beings worshipped by Shakta, Shaiva, and Vaishnava sects of Hinduism and by Buddhists. Although a significant medium for sacred sculpture, alongside stone, metal, and wood, terracruda images have received scant attention in scholarly literature. Surviving sculpture is rare, and scholars, preoccupied with sacred figures in durable and valuable mediums, have long overlooked fundamental distinctions between terracruda (air-dried clay), terracotta (low-fired clay), and stucco (gypsum or lime plaster that hardens through a chemical reaction). Despite the dearth of focused research on terracruda sculpture, much information can be gleaned on ancient and extant practices from early texts on image making and ritual practice, particularly Agama and Shilpashastra texts from southern India, legendary histories of dynasties, and stories of the gods (Luczanits 2004; Rai 2009; Robinson 1983; Varma 1970). Modern sources that bear on terracruda sculpture include colonial government publications and work by anthropologists, archaeologists, art historians, historians, and scholars of religion (see especially Guha-Thakurta 2015; Heierstad 2017; Kaur 2005; Robinson 1983; Sen 2016). To complement and extend these resources, the author researched contemporary practices in Bhutan, Maharashtra, and West Bengal.

The approach developed to terracruda sculpture gives paramount attention to the medium, airdried clay, by closely considering materials as active contributors to the lives of images. Drawing from recent scholarship that is reconceptualizing the nature of inanimate things and substances, materials are recognized not as passive stuff to be manipulated by human beings with intentions, but as participants whose intrinsic capabilities interact with other participating substances, objects, and animate actors. Insights into the nature

of materials by philosophers, anthropologists, and sociologists have destabilized long-established hierarchical distinctions between humans who act with intentionality, animals that behave instinctively, and inanimate things (Bennett 2010; Deleuze and Guattari 1987; Ingold 2011; Boivin 2008). Materials are established as not merely serviceable to human actors, but as "players in the world" (Bennett 2010, p. 3). This perspective provides a basis for considering clay and other substances in ways that reconfigure human relations with non-human animals, plants, and substances to more fully account for the complexities of ecosystems.

Among sculptural mediums, terracruda stands out for its radically transformational character—its penchant to move between states of hardness, plasticity, and fluidity. Hardened terracruda forms are forever prone to crumble into rubble or melt into slurry. In this region, especially in conjunction with goddess worship, clay is understood as a constituent of fecund earth, and clay-earth as a fundamental element of the cosmos, the support of all living things, at times personified as divine mother. The penchant of terracruda forms to disintegrate has been widely appreciated for harmonizing with cosmologies that conceptualize existence as cycles of creation and destruction or understand the material world as impermanent or illusory. Terracruda images perform the impermanence of things and the inevitable return to formlessness. Clay's capacity to repeatedly take form and disintegrate complements natural cycles, phases of the moon and the sequence of seasons that signal the time for worshipping a deity.

Clay modelers know that clays from different locales have distinctive capacities. Some are too coarse for refined modeling but fire successfully as earthenware. Others are excellent for fine modeling but crack or slump if fired. Some clays are too sticky to work with, or too sandy to retain a shape, or shrink and crack while drying. Although such faults can sometimes be successfully managed by blending clays and adding other substances, it is telling that clay modelers from Bengal who take commissions in distant locales often bring local modeling clays with them for refining the final form of a figure. Because clay can be kept plastic by maintaining moisture content, sculptors can repeatedly alter and rework postures, and their clients can take an unusually active role in the creative process, advocating alterations in iconography, posture, expression, and ornamentation. In eastern India and Bangladesh, ephemeral terracruda ritual images worshipped in periodic festivals engender perennial rounds of making, using, and disposing. Each occasion closes with the disintegration of images and initiates the need for new images. The ongoing requirement for images creates opportunities for sculptors to refine their practices and continually educate their eyes, hands, and creative artistry. Clay's flexible plastic character and air-dried forms' fragile ephemerality foster a temporal dynamic involving cyclicality, repetition, re-creation, and innovation.

### 3. The Rise of Terracruda Ritual Images from the Late 1500s

In the late 1500s, the worship of terracruda ritual images, long present in the region, increased in prominence. Following Mughal Emperor Akbar's victory over the ruling Sultanate of Bengal in 1575, provincial governors were placed in charge of the region. Despite being obliged to share revenues with new overlords, some zamindars succeeded in expanding their holdings and increasing their wealth, becoming in effect rulers of small kingdoms under Mughal suzerainty. Some among them who were devotees of the goddess instituted a new autumn Durga Puja that coincided with the harvest after the monsoon, the customary time for rulers to pursue military campaigns. Among them, three are considered founders of the autumn puja: Kangshanarayan of Tahirpur, now in Bangladesh, in 1583; Lakshmikanta Majumdar, whose family lands were to comprise a substantial part of Calcutta, in 1606; and Bhavananda Majumdar of Nadia, in 1610, ruler of a kingdom that would become the region's center for terracruda sculpture, (McDermott 2011, p. 14). Their celebrations of Durga Puja recast the festival as an occasion of royal splendor. Zamindars staged elaborate pujas in a *thakur dalan* (hall of worship), a purpose-built courtyard or pavilion adjacent to the palace that could accommodate large gatherings.



They installed an imposing terracruda image of the goddess on an elevated platform where Brahmin priests performed requisite rituals over the days of the festival and where devotees could experience auspicious visual contact *(darshan)* with the deity (Figure 6). Zamindars entertained many guests, fed numerous holy men, and offered sacrifices to the goddess. They sponsored recitations by poets and scholars and brought dancers, musicians, and dramatic troupes to perform. The events were aimed to please the goddess, impress and entertain their subjects, and boost their reputations in Bengal and beyond (Figure 7).

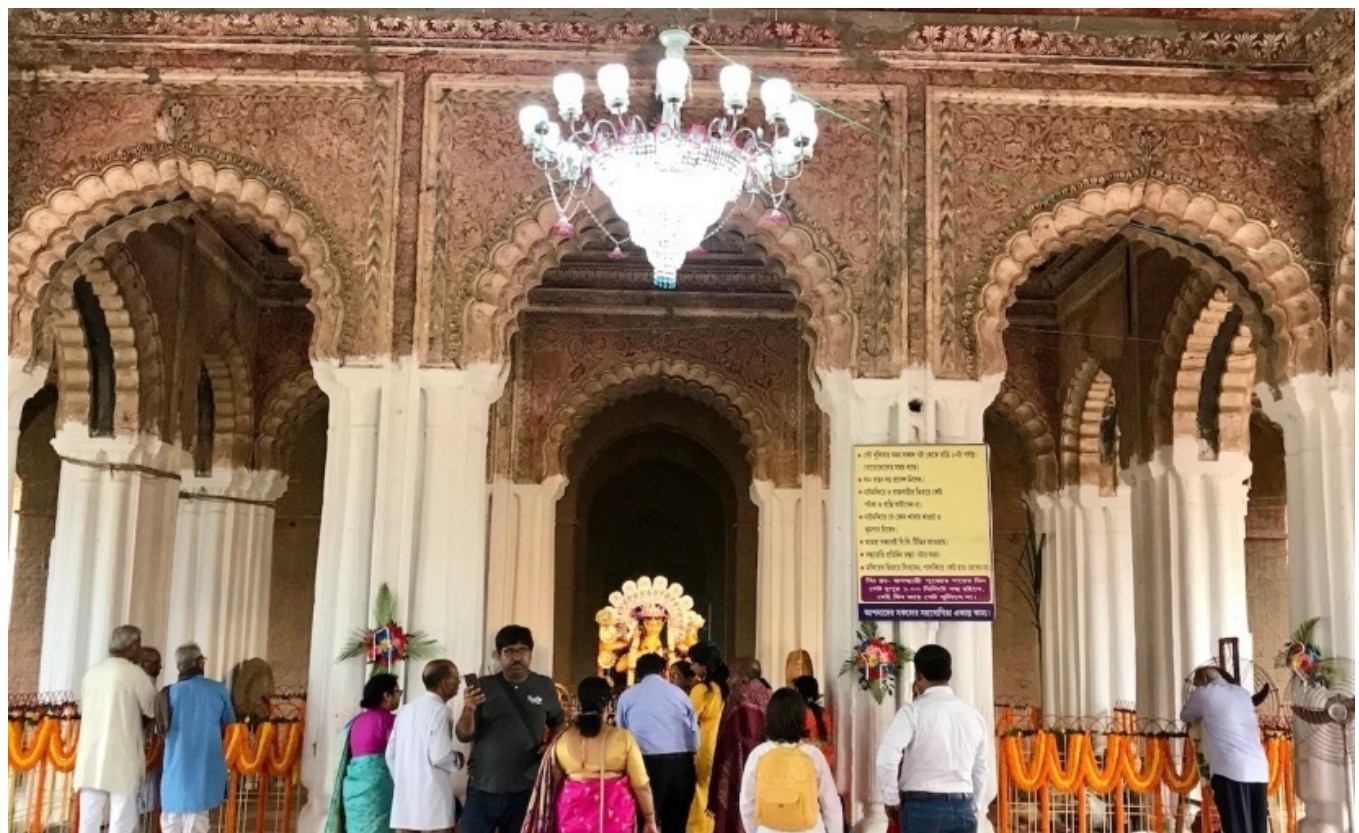

**Figure 6.** Visitors viewing the image of Jagaddhatri in the *natmandir* of the erstwhile ruling family of Nadia, at erected c. 1762 by Raja Krishnachandra Ray of Nadia. Courtesy Blog of Kinjal Bose.

While the magnificence of the autumn Durga Puja was new, worshipping the goddess in a terracruda representation had older roots. Bhavadeva Bhatta, an authority on sacred scriptures and moral conduct active around 1150 CE, cited earlier Sanskrit texts that refer to worshipping the goddess in terracruda images (Sastri 1960, p. 268). Other learned commentators alluded to the rationale for terracruda, most notably Krittibas Ojha, who composed a Bengali version of the epic *Ramayana* in the 1400s. He interpolated an episode absent from other renditions that describes the hero Rama on the brink of his autumn battle with the formidable Ravana. Rama appeals for support to the goddess's martial manifestation as Durga and the goddess responds, directing him to worship her as a ten-armed, weapon-wielding earthen (terracruda) image. Rama did as instructed, successfully enlisting the goddess's awesome powers for a victorious outcome (Nicholas 2013, pp. 55–56). In another episode that parallels Rama's puja to the goddess in a terracruda image, the *Devi Mahatmya* (Glory of the Goddess) compiled in the mid-first millennium CE articulates the goddess's role as powerful protector, supporter, and sustainer. The goddess instructs a deposed king and a once-prosperous merchant seeking her beneficence to demonstrate their worthiness by performing austerities and worshipping her in an earthen image. Once assured of their sincerity and devotion, the goddess grants their wishes (Coburn 1991, p. 83, verse 13.7). Another passage of the *Devi Mahatmya* describes the goddess's identity with fecund earth:

"You alone are the sustaining power of the world, for you abide in the form of the earth. By you, who exist in the form of water, all this universe prospers, O Devi of unsurpassable strength" (Nelson 2003, p. 188, verse 11.4). In modern Bengal, some practices reflect the continuing salience of clay-earth's potency. In rural areas, one who gathers clay to make a ritual image may first bow to the ground to honor earth's divinity and identity with the goddess (Nicholas 2016, p. 57). In a similar vein, contemporary terracruda sculptors in Bangladesh assert that firing an image destroys the lifeforce that clay harbors (Glassie 1997, pp. 330, 346).

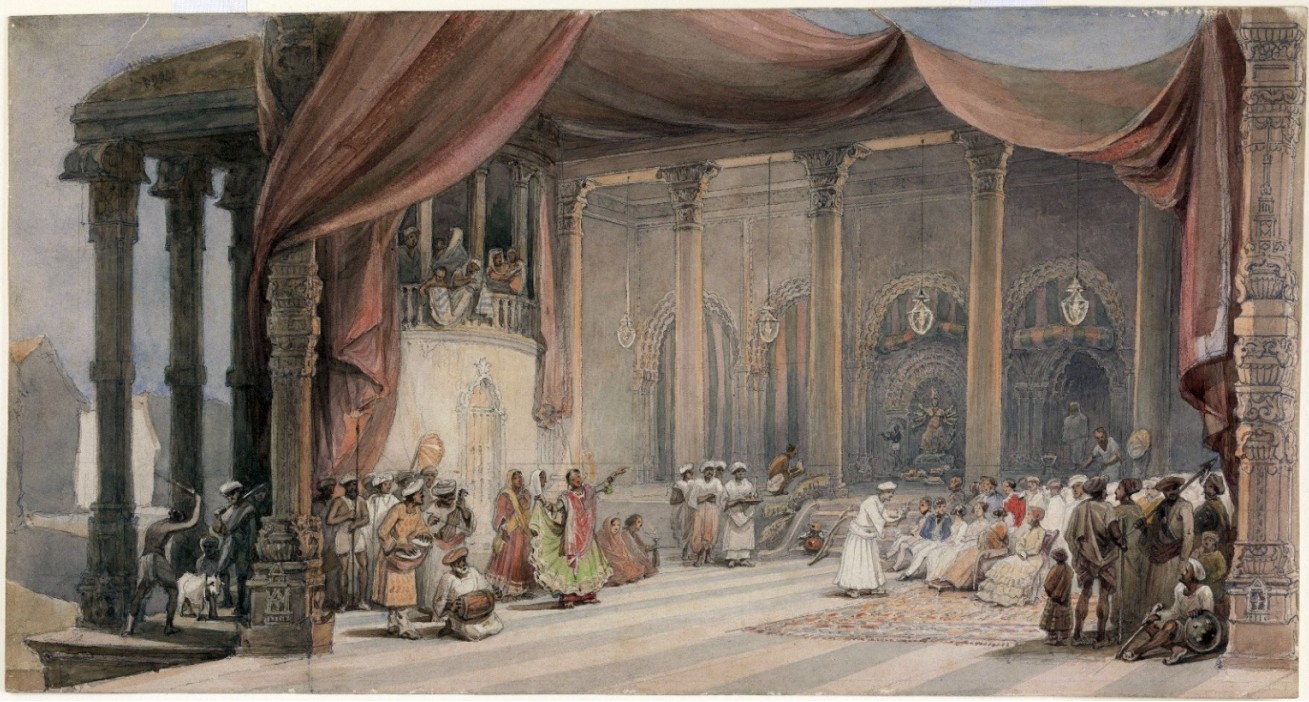

**Figure 7.** William Princep (1794–1874), "Europeans Being Entertained by Dancers and Musicians in a Splendid Indian house in Calcutta during Durga Puja." Watercolor, 22.9 cm × 43.5 cm. Calcutta, 1830–1840. Collection British Library, Wikimedia Commons.

For the zamindars who established autumn Durga Puja as a principal event of the ritual calendar, the parallel with the Bengali version of the *Ramayana* was especially salient as it demonstrated the goddess's support for pious, worthy rulers, protectors of their subjects. To mount their elaborate courtly pujas, zamindars relied on skilled local modelers and supplies of fine modeling clays that enabled sculptors to create ritual images of suitable splendor. Some zamindars commissioned images that emphasized Durga's martial prowess, depicting her as a warrior in a trouser-like garment with a jewel-encrusted breastplate, brandishing her weapons and standing with her right leg on the back of her mount, a *ghoḍa simha* (horse-lion), and her left leg on the shoulder of the vanquished demon, Mahishasura (Chatterjee 2013, p. 1467). In another innovation unique to the region, zamindars commissioned images incorporating the deities Lakshmi, Saraswati, Ganesh and Kartik as Durga's four children standing at her side. The new visualization of the goddess was likely a response to the surge of Vaishnava devotional worship emphasizing intimate emotional ties between worshipper and god. The presence of Durga's children in the image, and the region-wide interpretation of her autumnal puja as a much anticipated annual return from her consort's mountain abode to her natal home, emphasized the goddess's nurturing, maternal, filial, protective aspects and linked the puja to an older spring festival that celebrated Durga's identity with fecund earth, abundant crops, flourishing herds and prospering families, the resources on which zamindars and their subjects relied (Kinsley 1986, pp. 95, 113–14; McDermott 2011, p. 6; Nicholas 2013, pp. 26, 44–45). At the close of the festival, the goddess's presence

departed. Worshippers lamented her absence as they conducted the terracruda figure in procession through the streets for all to see. The procession ended at the waterside where the image was immersed to disintegrate and return to the earth, articulating the ongoing cycle of making, worshipping, disposing, and remaking, and instigating devotees' anticipation of the goddess's return as well as speculation on how she would be visualized and what the experience of visual communion (darshan) would bring next time. In practice, the ongoing cycle of making and disposing became instrumental to an expansion of the festival cycle in a manner that responded to political, economic, and social developments in the region. Clay's superb plasticity and penchant to disintegrate positioned it as an engine of innovation, and the establishment of cyclical festivals centered on temporary terracruda images ensured the continuing evolution of the art.

### 4. The Elevation of Terracruda Sculpture in the Eighteenth Century

In the eighteenth century, Raja Krishnachandra Ray, zamindar of Nadia (r.1728 to 1782), was among the most powerful local rulers in the region. Under the Mughal provincial government, he expanded Nadia to 3000 square miles. Raja Krishnachandra became renowned not only for the size and wealth of his kingdom, but for his energetic support of a broad spectrum of arts and learning, embracing Sanskrit literature, religious scripture and practice, classical music, dance, architecture, terracruda sculpture, colloquial poetry, antics of saucy court jesters, and boisterous, bawdy rhyming competitions (McDermott 2001, p. 20). Nadia became the cultural center of Bengal, and Raja Krishnachandra was likened to Vikramaditya of Ujjain, a legendary just king and patron of the arts and learning (Anonymous 1850, pp. 124, 130–31). Yet, Raja Krishnachandra ruled during challenging times. In the 1740s, armies of the expanding Maratha Empire launched a decade of military incursions, raiding towns and villages, extracting tribute from the Mughal governor and wreaking havoc on the lives of ordinary people. In the 1750s and 1760s, the British East India Company, headquartered just downriver in Calcutta, transformed from a trading corporation into a military and political power. After victory at the Battle of Plassey (Palashi) in 1757, the Company secured tax collecting rights from the provincial Mughal government. With the defeat of the Mughal army at Buxar in 1764, the East India Company became the ruling power in Bengal. A series of crop failures beginning in 1768, exacerbated by the Company's system of tax farming and by merchants profiteering from scarce supplies, precipitated catastrophic famine and heightened vulnerability to epidemic smallpox. Across the region, millions died. As their territories suffered military incursions, famine, disease, and growing British power, Raja Krishnachandra and other zamindars struggled to maintain and protect their kingdoms. They turned to the goddess for her support of their regimes and their subjects, and to bolster their stature among the region's elites.

Over his long reign, Raja Krishnachandra was a leading advocate for devotion to the goddess and for enlisting terracruda sculpture to spread goddess worship. His capital in Krishnanagar, situated on the banks of the Jalangi River, lay adjacent to unusually rich deposits of excellent modelling clays that balanced sandy, loamy, and sticky qualities providing sculptors with a nimble plastic mass that did not cling to the hand, maintained, details of form, and dried to hardness without serious cracking, crumbling or loss of detail (Chakravarti 1985, pp. 3–4). Fine modeling clays facilitated the creation of sacred images that could simultaneously honor iconographic form and accommodate sensitive alterations. In Krishnachandra's era mritshilpis benefited from clay's capabilities which supported their endeavors to create the larger more captivating and impressive images that zamindars required for royal pujas, enabled them to devise the composite forms that integrated Durga's children in a harmonious arrangement under a single arch, and facilitated making adjustments in the goddess's expression and posture to simultaneously emphasize her martial prowess and her nurturing, protective nature.

Raja Krishnachandra is credited with nurturing the growth of terracruda sculpture and developing Krishnanagar into the leading center for terracruda sculpture, known for the "best clay images in Bengal" (Chakravarti 1985, p. 4; Ward 1811, vol. 3, p. 364). He offered

practitioners grants of land to settle in his capital and in other towns in his territory. He rewarded *mritshilpis*, clay sculptors, for outstanding work. Mritshilpis in Krishnanagar refer to a forebear, Gobindo Pal, whose fine representations of the goddess were honored with a gift of land in a nearby town (author's conversation with local historian Mohit Roy in 1996). In Nabadwip, during Ras Yatra, when Vaishnava devotees celebrated the divine dance of Krishna and Radha, Raja Krishnachandra encouraged competing pujas to the goddess. He supported the celebration by arranging for the terracruda images to be presented at the palace on the way to the river for immersion. The most outstanding representations of the goddess received awards (Chakravarti 1985, p. 10). Raja Krishnachandra's grandson and successor, Girishchandra Ray, is said to have continued the practice. Even in the 1980s, terracruda images of Durga, Kali, and Jagaddhatri were paraded to the palace to be viewed by descendants of the erstwhile royal family, a vestige of the part that Nadia rajas once played as patrons of the festival and arbiters of artistic quality (Robinson 1983, p. 179).

Raja Krishnachandra is also known for expanding the cycle of pujas. By 1800, he had become legendary for "impart[ing] celebrity" to Durga Puja, elevating the autumn festival as the region's premier celebration, a position maintained over the ensuing centuries (Anonymous 1830, no. 1; p. 1990). Raja Krishnachandra and his successors were also devotees of Kali, remembered for exhorting "all the people of . . . the country over which they had a nominal authority to perform the worship of Kalēē, and threaten[ing] every offender with the severest penalties on non-compliance." It was said that because of this decree, in the early 1800s, more than ten thousand households worshipped Kali on the night of her puja (Ward 1811, vol. 3, p. 191). While many households installed paintings on paper in domestic shrines and some, including the ruling family's, worshipped stone images, polychrome terracruda images became increasingly prevalent. Raja Krishnachandra is also recognized for instigating new or revitalized annual pujas centered on terracruda images of other manifestations of the goddess. The raja's court poet Bharatchandra Ray, in his *Annandamangal* (c. 1751), related that Nawab Alivardi Khan, Mughal viceroy of Bengal (1740–1756), imprisoned Krishnachandra for refusing a demand for tribute. While Krishnachandra was held captive, Annapurna, goddess of sustenance, appeared to him in a dream and directed him to establish her annual worship and commission his court poet to compose her 'auspicious song.' Annapurna Puja, centered on terracruda images, became a signal occasion of the spring season enlisting the goddess to assure the productivity of crops and animals vital to the agricultural communities that were the foundation of zamindars' power (Curley 2008, p. 207; Robinson 1983, pp. 143–44). An almost identical account, with a shaky historical basis, is widely cited as the origin of a new annual Jagaddhatri Puja (Lyons 2009; McDermott 2009) (Figure 8).

Documentary sources suggest that Jagaddhatri Puja is more likely to have been established during the reign of Krishnachandra's grandson, Girishchandra (r. 1802–1841). However, in the popular version, Raja Krishnachandra, again incarcerated by the Mughal governor, receives the goddess in a dream. She consoles him for missing Durga Puja and instructs him to worship her as Jagaddhatri in a terracruda image that depicts her mounted on her lion with weapons in her four hands, vanquishing a world-threatening elephant demon. A mid-nineteenth century account notes that this new puja rapidly took hold. "The Goddess Jagaddhatri in the present form, was unknown to the Hindoos who died a few centuries ago . . . . She is Doorga in a different form, sitting on the back of a lion who also rides on an elephant and tears it to pieces" (Gangooly 1860, p. 155). Such stories swirling around Raja Krishnachandra have obscured his actual contributions, and credit for the modern florescence of a terracruda sculpture settled around him in an almost mythical aura (Bordeaux 2015, pp. 5–6, 161; Lyons 2009, p. 262). Over time, other deities were incorporated into the annual festival cycle and worshipped in temporary terracruda images, including the goddesses Lakshmi, bestower of wealth and good fortune; Sarasvati, presiding over learning and the arts; Shitala, goddess of smallpox; and Manasha, goddess of snakes—as well as male deities, especially Kartik, commander of the gods' armies; and Vishvakarma, architect of the gods and patron of skilled work. Even the Vaishnava festival

of Rash Yatra was elaborated with terracruda images of Krishna, his beloved Radha, and the Gopis, female cow herders who are their companions.

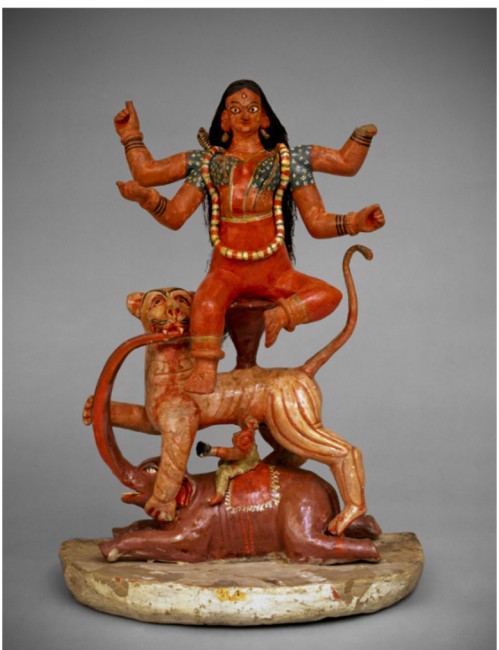

**Figure 8.** Polychrome terracruda, Calcutta c. 1815; height 43.5 cm. Courtesy Peabody Essex Museum.

The expansion of the cycle of festivals was facilitated by clay's supple and enduring plasticity that enabled mritshilpis to refine or alter details of established forms and devise new forms with a latitude unmatched by other materials. Mritshilpis elaborated an existing technique using bamboo or wood padded with vegetal matter for armatures. They created larger flexible supports from rice straw or native *kusha* grass by bunching, twisting, and tying bundles into the shape of torsos and limbs, a method that enabled them to produce armatures more than ten meters high over which they could layer masses of moist clay. Working with clay particles' penchant to join together, they smoothed and compressed the layers of clay into an integrated mass (Ward 1811, vol. 3, p. 364) (Figure 9).

By the end of Krishnachandra's reign, his holdings and those of other zamindars were eroding under East India Company rule. The Company's inflexible requirements forfeited zamindars' rights to collect taxes if payments were overdue. In their place, outside investors were empowered as tax-farming landlords, diminishing zamindars' resources and undermining their standing (McDermott 2011, pp. 18–19). Their declining powers and fortunes undercut zamindars' ability to support lavish pujas and generous patronage of temples, theologians, poets, painters, and sculptors. At the same time, the new colonial capital of Calcutta was emerging as a cultural center. The city, created in 1690 from the lands of three villages, was shaped and dominated by the British from its inception. A century after its founding, as power in the region shifted decisively from Mughal suzerainty and local zamindari estates to East India Company control, the British set about remaking Calcutta from a fortified trading center into a fitting capital for Bengal and British territories in India. Imposing buildings were erected that favored neoclassical style with its imperial allusions. British officials and successful businessmen acquired fine city residences. Local entrepreneurs who facilitated the British elite amassed fortunes as bankers and commercial agents and erected their own palatial mansions. Landed gentry rooted in the countryside also kept grand residences in the city to maintain their standing in the new regime. Ordinary laborers, merchants, and skilled makers, including masons, painters, metal smiths, potters, carpenters, and terracruda sculptors came from near and far for opportunities to meet the city's growing needs—and some among them prospered (Banerjee 1989, p. 55). Under colonial authority, Calcutta took shape as a heterogeneous and cosmopolitan center.

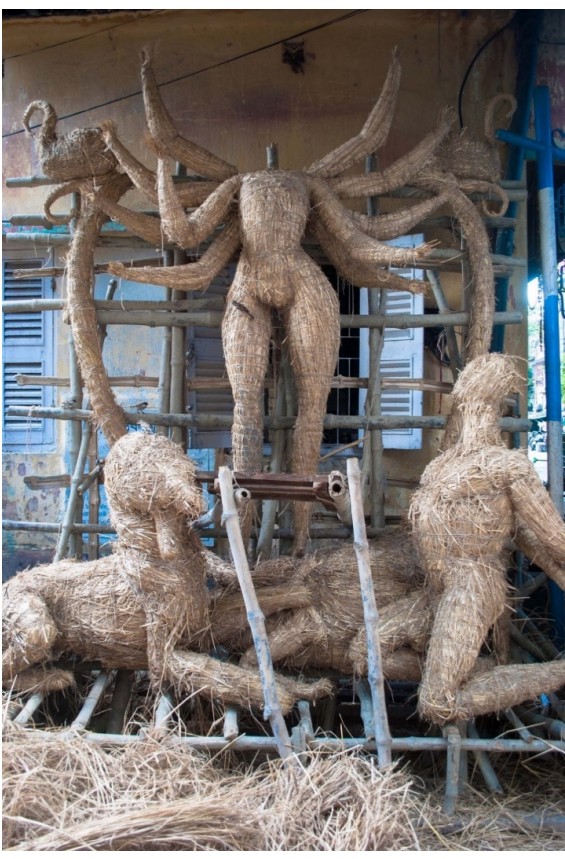

**Figure 9.** Straw armature for Durga, her lion vehicle and Mahishasura, approx. height, 4 m. Kumortuli, Kolkata 2013. Photograph by Sujay25, Wikimedia Commons.

## 5. Terracruda as an Engine of Innovation in Colonial Bengal

The shifts in power and social composition soon reverberated in the staging of periodic religious festivals. The new Bengali elite mounted grand occasions simulating the splendor of zamindars' celebrations and serving as occasions to impress foreign guests and vie for locals' esteem (see Figure 7). Raja Nabakrishna Deb (1737–1797), for example, born to a family of high rank and modest fortune, carefully educated to succeed in the new environment, became wealthy as a facilitator of East India Company endeavors. In 1757, he persuaded the British general Robert Clive to celebrate the victory at Plassey as part of the city's first elaborate Durga Puja to be held at the Deb mansion. To ensure having the finest terracruda image of the goddess for his family's thakur dalan, Deb enlisted a mritshilpi from Krishnanagar (Heierstad 2017, p. 60). Other elite Bengalis followed his example, bringing Krishnanagar's leading sculptors to make ritual images for pujas at their residences. Eventually, mritshilpis from Krishnanagar and other towns in Nadia settled in Kumartuli, the potters' quarter, creating a new center for terracruda sculpture.

In the countryside, as zamindars' wealth and power dwindled, well-off households, typically Brahmins, pooled resources to organize pujas in their communities. Around 1790 in the Nadia town of Guptipara, several Brahmins joined together to organize an autumn puja for the goddess Jagaddhatri. To augment their resources, they sent representatives to surrounding settlements and far-flung towns to prod others to contribute. They gathered sufficient backing to mount a celebration with some of the splendor and excitement of a royal puja, hosting a weeklong gathering of Brahmins for an occasion that would be remembered for its festive atmosphere. Such community-backed pujas came to be known as *barowari* (literally, twelve friends) and spread readily, fostering local control and public involvement (Long 1820, p. 129). To attract worshippers and admirers from near and far, organizers selected prominent locations with space for a temporary pavilion to shelter the deity's image and an adjacent area with room for worshippers to gather and

performances to be held. The *pandals* (temporary pavilions) were usually constructed with bamboo supports and cloth coverings. Inside on an elevated platform, they installed a large and imposing terracruda ritual image typically two meters high (Ward 1811, vol. 3, p. 364). (See Figure 3). They enlisted priests to perform the rituals, and sponsored plays, performances by singers and dancers, and recitations from the epics. An observer in the 1850s wrote, " . . . a party of young men tax each family once a year and dispose of the money thus realized in the 'worship of twelve friends' . . . A splendid image is erected in the public place of the town, and music and theatrical performances are held before it" (Gangooly 1860, pp. 177–78).

Community pujas took root and spread to other festivals, especially in the diverse and fluid social milieu of colonial Calcutta (Bhattacharya 2007, p. 946). In the mid-nineteenth century, Kaliprasanna Sinha, a renowned Bengali satirist with a keen eye for the changing ambiance, noted: "Earlier Durga puja used to be celebrated only in the houses of rajas and rich people. However, nowadays, even two-bit oil pressers could be seen carrying home an idol of Durga for worship!" (Sinha [1862] 2008, p. 152). Sinha delighted in the pujas patronized by "loan sharks, wholesalers, shopkeepers and vegetable sellers..." Noting the rise of community-organized pujas as a means for gaining local recognition and a new emphasis on attracting visitors and supporters, he wryly observed, "some fun-loving and foppish merchant . . . acts as the chief patron of the puja committee . . . to collect subscriptions and make arrangements for *shong* shows" (Sinha [1862] 2008, p. 26). The terracruda *shongs*, likely spinoffs of pantomime actors' performances also known as *shongs*, represented legendary events, illustrated proverbs, and lampooned local elites (Banerjee 1989, pp. 14–15, 39–44). Sinha remarked on the ambiance: "There was fierce competition between the various community pujas. Santipurwallas [residents of Santipur] once spent five lakhs [INR. 500,000] on a community puja . . . The idol installed was sixty feet high, so it had to be cut into pieces on the day of immersion!" (Sinha [1862] 2008, p. 32). On the final day, organizers publicly escorted the images in procession to immersion, vying for spectators' approbation. "The subject in the city that day was: 'Whose idol is the best?' "Whose pageantry is the grandest?' and 'Whose arrangements are the finest?'" (Sinha [1862] 2008, p. 163).

The more fluid social order that prevailed under East India Company rule gave reign to the nimble plasticity of clay. New elites, prospering small-scale merchants and skilled workers could commission distinctive ritual images for their pandals and display other terracruda figures in shong tableaux adjacent to the pandals. The increasing number and diversity of terracruda figures in demand, and the competitive ambiance of the festivals, shifted the dynamic of mritshilpis' practices, putting a premium on ingenuity in devising striking ritual images and depicting characters from the past and contemporary society. Kaliprasanna Sinha described a remarkable image of Jagaddhatri: "The community puja idol was about twenty feet high. It was encircled by figures of Highland soldiers on horseback...The goddess Jagaddhatri was seated in the middle . . . .The goddess looked like a memsahib [European woman]; her complexion and build were that of a pukka [proper] Jew and Armenian! Brahma, Vishnu, Maheshwara and Indra were singing hymns to her with folded hands. Little British fairies were blowing horns on top of the idol; they had royal insignia in their hands. In the middle there was the Queen's *unicorn* and *crest*" (Sinha [1862] 2008, p. 44). Most such playful take-offs from tradition occurred not in ritual images of the goddess, but in the shongs intended to entertain. Sinha describes several of these including a shong of Mr. Upstart, who "had a beautiful pinkish complexion and wore his hair [Prince] *Albert fashion* . . . one would take him to be of royal descent, but enquiry would reveal that his grandfather was a petty weaver" (Sinha [1862] 2008, p. 44). Ritual images changed more slowly, first affecting subsidiary elements of the images such as the "little British fairies" and the fashion-conscious representations of Kartik in Durga Puja images. In the early 1800s, the goddess's son Kartik, placed at her lower left, was typically depicted as a dandy, "a handsome young man, with long and curled hair hanging down to his neck, a thin trace of a moustache, dressed in superfine dhoti, a light plaited scarf thrown around his neck and wearing a pair of gold-embroidered shoes . . . " In the late 1800s,

he "sport[ed] the [Prince] Albert fashion of hair style, . . . English shoes . . . and, a jacket" (Banerjee 1989, p. 129). British visual culture, accessible through, prints, book illustrations, statuary, and paintings, was appropriated and manipulated to suit the festival spirit.

Mritshilpis had begun exploring European naturalistic figuration around 1800 for shongs and subsidiary figures in ritual images. They also received commissions for portraits and for ornamental work including imitations of famous European statues for mansions of the nouveau riche. Rajendra Mullick, who built a palatial residence in the late 1830s known as "the Marble Palace", commissioned statues of heroes from the Ramayana, most likely in terracruda, to display in the thakur dalan, where ritual images were installed during periodic festivals (Chaliha and Gupta 1990, p. 178). Foreigners, including British and American sojourners in Calcutta, commissioned depictions of potters, priests, musicians, and other locals at their callings, as mementos (Bean 2001, pp. 183–88). In the mid-1800s, three Bengali *banians* (commercial agents) for the American trade, had their own life-size terracruda portraits made to send to the East Indian Marine Society Museum in Salem, Massachusetts. One of them, Rajinder Dutt (Figure 10), was an avid patron of terracruda sculpture. During Rash Yatra and other festivals, he placed figures in the lane outside the family residence depicting characters from works by William Shakespeare and Sir Walter Scott, as well as leaders of the American and French revolutions (Bean 2001, pp. 218–21). However, the liveliest venues for innovative work were the terracruda shong tableaux displayed at community pandals during festivals. Kaliprasanna Sinha critiqued them, praising some as clever caricatures and berating others for clumsy portrayals. He reveled in a send-up of a hypocritical holy man, "Mr. Holier-than-thou [whose] body... bent with age...was clicking the beads of a rosary and ogling women of respectable families!" However, at another pandal, the depiction of legendary ruler Vikramaditya was so wretched that the king resembled an "opium agent" and all his courtiers had the same disheveled look, "like a pack of poor Brahmins . . . ." (Sinha [1862] 2008, pp. 40, 43–44).

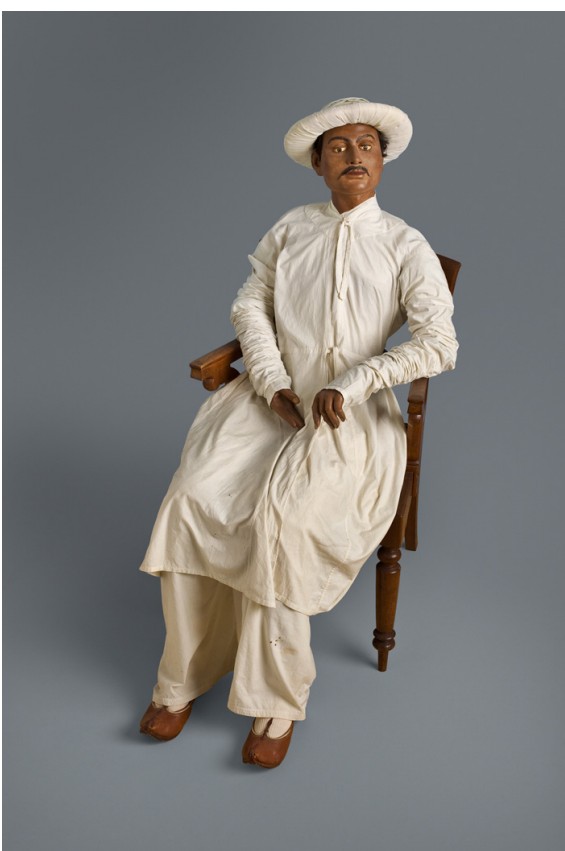

**Figure 10.** Sri Ram Pal (attributed), portrait of Rajinder Dutt, polychrome terracruda, cloth, hair, chair; life size, c. 1849. Courtesy Peabody Essex Museum.

The shongs freely incorporated western naturalism to portray heroes, mock hypocrites, and playfully indigenize western cultural icons. Through the 1800s, mritshilpis, facilitated by their plastic medium, were among the vanguard of local artists adapting elements of western visual culture into their widely seen work, contributing to the development of a colonial modern visuality. The penchant of terracruda to disintegrate, and the relatively modest expense of commissioning new figures, ensured that not only ritual images, which were made to be short-lived, but also terracruda shongs and decorative statuary would be replaced every few years, creating new opportunities for inventive, imaginative work.

## 6. Postcolonial Bengal

In the early 1900s, India's nationalist movement gained traction in Bengal when the colonial regime's division of the province into two parts galvanized political resistance. Activists gained broad support for a campaign that encouraged using *swadeshi* (own country) goods and boycotting British imports. A pamphlet in circulation *Bharater Sarbajanin Durgotsav* (The Durga Festival for all the People of Bharat/India) positioned the Puja as an event to gather support for the nationalist cause. In Calcutta, a few Durga Puja organizers, building on innovative practices of the nineteenth century, created pandals representing the goddess in the guise of Bharat Mata (Mother India) (Bhattacharya 2007, p. 955; McDermott 2011, pp. 57–64). Once again, clay's plasticity and the ephemerality of terracruda images facilitated mritshilpis and puja sponsors in refashioning the Puja to bolster political aims, in much the same way that Bengali zamindars in the 1600s favored martial images of Durga to portray her as the militant supporter of righteous rulers.

With the allied victory in World War I, leaders of the nationalist movement expected the colonial regime to initiate steps towards self-rule in recognition of the service provided by Indian troops. Instead, new restrictions were imposed. In response, Mahatma Gandhi and the Indian National Congress launched a non-cooperation movement that roused participation. As the struggle for Indian independence grew into a mass movement, in Bengal nationalist leaders pressed for Durga Puja, the premier festival, to be a more public occasion with neighborhood pandals accessible to all as *sarbajanin* ('universal, for all people') including worshippers, activists, curiosity seekers, and festive revelers rambling through city neighborhoods to view the splendors of pandals and terracruda images of Durga (McDermott 2011, p. 65). Puja organizers responded to this new climate of openness, initiating a popular trend that favored innovation, even in depictions of deities.

Among mritshilpis, Gopeshwar Pal took on an outsized role in stimulating the transformations of ritual imagery. He became famous for jettisoning iconographic conventions and abandoning the established composition that placed Durga at the center posed in triumph over the demon Mahishasura with her children beside her, all assembled under an arch. Gopeshwar began in the 1920s, revolutionizing the representation of deities. He abandoned the arch providing space for each figure. He enlisted the clay's plasticity and the flexibility of straw armatures to animate the figures with naturalistic poses and expressions, providing each with distinctive character (Guha-Thakurta 2015, pp. 162, 167; Heierstad 2017, p. 100; McDermott 2011, pp. 110–13; Sen 2016, pp. 162–64) (Figure 11).

Gopeshwar Pal's innovations enhanced the public appeal of Durga images. Naturalistic portrayals of deities and dramatic departures from established iconography had caught on decades earlier in the 1870s, when painters at the Calcutta Art Studio, educated in the colonial curriculum of the government art school, translated the art style of the ruling class to accommodate naturalistic portrayals of the gods that were simultaneously awesome, approachably lifelike, and emotionally accessible. Their paintings were used as templates for the mass production of inexpensive chromolithographs of the gods, part of a trend across India that saw art school educated painters producing prototypes infused with naturalism that found growing clienteles (Guha-Thakurta 1992; Jain 2007; Kapur 1993; Mitter 1994; Pinney 2004). Many devotees installed these chromolithographs in domestic shrines for private worship. However, priests and organizers of community pujas resisted such novel depictions of deities in the terracruda images that were central to the cycle of

public pujas. Until the early twentieth century, mritshilpis and trend setting puja organizers restricted novelty in ritual images to minor alterations and pursued innovation principally in the shong figures that were displayed nearby.

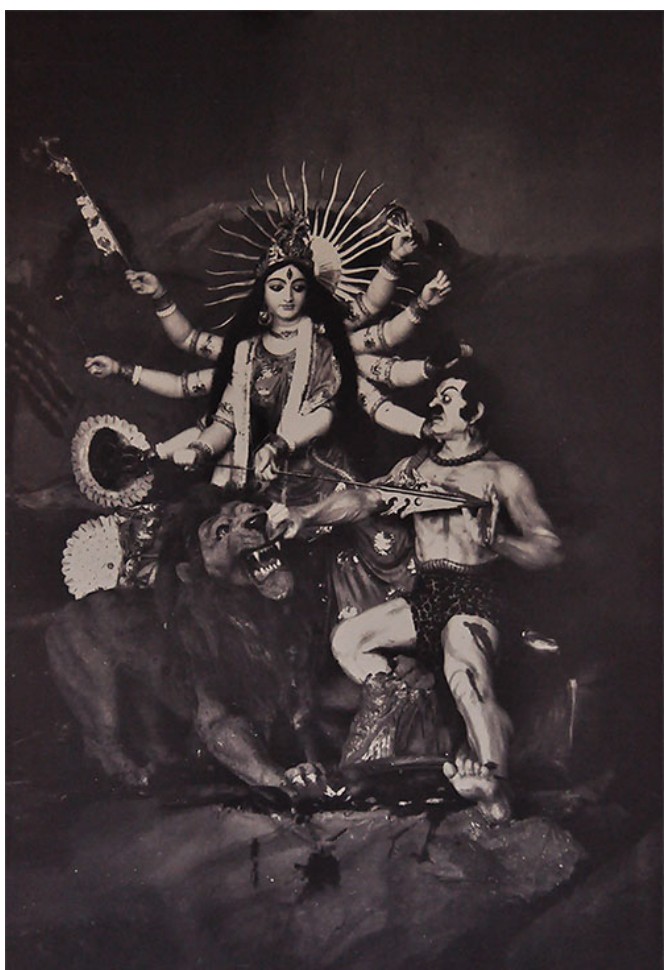

**Figure 11.** Gopeshwar Pal, Durga image, terracruda and various media, approx. 4 m. Calcutta, 1940s. Photograph from a Calcutta newspaper, posted on Twitter, 2020 (Swati Moitra @swatiatrest).

Because mritshilpis had been working with European naturalistic realism in fashioning terracruda festival shongs, ornamental statuary, and portraits since the end of the eighteenth century, Gopeshwar Pal and other clay sculptors were well prepared to upend the strictures on terracruda ritual images. Gopeshwar had apprenticed with Jadunath Pal (1834–1924), a leading exponent of the Krishnanagar naturalistic style who produced many of the life-size realistic figures representing the inhabitants of India that were displayed in ethnological sections of late nineteenth century international exhibitions (Bean 2017). In the 1920s, Gopeshwar was invited to participate in London's Festival of Empire (1924) and took the opportunity to spend months in Italy gaining firsthand experience of Classical Greek and Roman sculpture and visiting workshops of contemporary Italian sculptors. His sojourn abroad invigorated his practice and stimulated his interest in deploying figural naturalism to enliven portrayals of the gods. When the advancing freedom movement in the 1920s emboldened puja organizers to commission revolutionary Durga images, Gopeshwar was ready to oblige (Guha-Thakurta 2015, pp. 167–68; McDermott 2011, pp. 110–11). Notably, it was the Kumartuli puja association in the mritshilpis' neighborhood where Gopeshwar lived and worked whose members had the confidence to move the portrayals of deities in new directions. Other puja associations and mritshilpis soon engaged with the new innovative spirit embarking on their own inventive paths. In the 1930s, N. C. (Nitai

Chandra) Pal in the swadeshi spirit enlisted styles of figuration from fourth-century Gupta sculpture and fifth-century murals at the Ajanta caves and contemporized Durga puja images with stylistic allusions to the work of prominent Bengali modernists Jamini Roy and Nandalal Bose (McDermott 2011, pp. 113, 174).

By the 1960s, puja organizers and mritshilpis could choose from a wide range of stylistic options for the ritual image of Durga, from old-style large-eyed, stiff-postured figures tightly grouped under an arch to representations of Durga resembling well-known film stars. Mritshilpis vied to "produc[e] newer and more artistic images and to draw... larger crowds of spectators" and the evenings became a time for people to visit pandals to view the images at least as much for their artistry as for darshan (Sarma 1969, p. 587). The popular appetite for innovative terracruda images and pandals reinforced puja organizers and mritshilpis' efforts to devise novel forms, creating Durga images "to fit almost every whim of fashion" (McDermott 2011, p. 114). As stylistic departures became more acceptable, mritshilpis' visibility rose with recognition for their individual contributions. Some puja organizers initiated the now common practice of placing plaques near ritual images that identified the maker as "the artist" (Sarma 1969, p. 588). At the same time, more mritshilpis extended their practices with forays into the burgeoning realm of civic statuary as the demand rose for portraits of national heroes, philanthropists, and educators. These works were typically originated in terracruda but cast in more durable plaster of Paris or cement, or replicated in marble, for long-term public display. Gopeshwar Pal was also a pioneer of this work. His life-size marble portrait of Ramakrishna Paramahamsa is enshrined in the temple at Belur Math and his larger-than-life portrait of Sir Rajendranath Mookerjee remains on the grounds of the Victoria Memorial (Guha-Thakurta 2015, p. 168) (Figure 12).

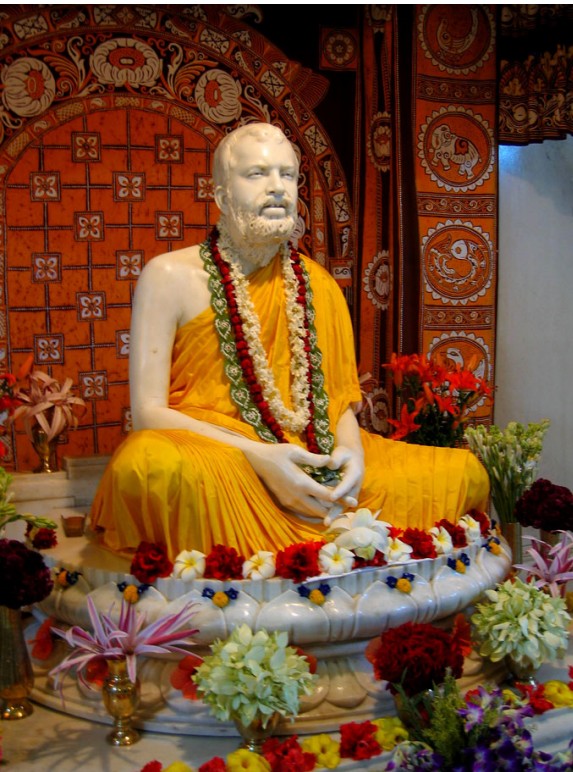

**Figure 12.** Gopeshwar Pal, Ramakrishna, Paramahamsa, marble life-size, c. 1930, at Belur Math, headquarters of Ramakrishna Mission, Howrah, West Bengal. Wikimedia Commons.

## 7. Conclusions/Discussion

In 1991, to avert the specter of sovereign default, the government of India loosened regulations in its state-controlled economy, easing the way for the rapid growth of private enterprise and international investment. These developments once again precipitated

dramatic changes in the infrastructure for public pujas. Game-changing new materials became increasingly accessible, including thermocol (polystyrene foam) that could be carved and painted, fiberglass that could be cast from terracruda originals to create inexpensive, lightweight, durable figures, and LED lights that enabled spectacular, animated illuminations (Banerjee 2017). Funding sources for public pujas proliferated as private corporations welcomed opportunities to garner good will and advertise their products with sponsorships of spectacularly scaled-up puja installations. Puja associations, pandal builders, image-makers and decorators created ever-more magnificent projects, especially for Durga Puja but spilling across the festival calendar.

In the twenty-first century, ambitious puja associations devised a new apparatus for developing Durga Puja installations, dubbed 'theme pujas.' Puja organizers determine an overarching concept for pandal installation and engage a specialist in the new role of 'designer' to take charge. The puja designer takes on the responsibility for developing, coordinating, and implementing the appearance of the pandal, the ritual image, and the decor. This individual might be recruited from the ranks of pandal designers, decorators, or mritshilpis, but art school graduates predominate. Some model the ritual images themselves; others engage mritshilpis to create the images (Guha-Thakurta 2015, pp. 199–246). When priests deem an image too unconventional for worship, puja organizers commission a mritshilpi to make a smaller, more traditional terracruda image to place nearby as the focus for the requisite rituals and immersion.

Some artistically ambitious puja designers who also produce work shown in art galleries are accelerating the dismantling of the division between 'art' and 'artisanry' that was institutionalized during British rule. The distinction had been problematic from the outset. In 1851, Krishnanagar mritshilpi Sri Ram Pal's lively lifelike figurines of Bengalis practicing their vocations were much admired and garnered an award at the *Great Exhibition* in London's Crystal Palace. Sri Ram Pal received no explicit recognition; he was considered a skilled worker, not an 'artist.' The award was made to the East India Company for providing the figurines. The figurines first entered the exhibition in the class for 'fine art,' but by the time the awards were announced, they had been reassigned to a new class for educational models. All the works that remained in the 'fine art' class were produced by European makers, including some models of social types much like Sri Ram Pal's figurines: the revised classification of 'fine art' was secured as exclusively the domain of Europeans working in an aesthetic lineage emanating from Classical Greek and Roman traditions (Great Exhibition 1852, p. 648).

In the 1930s and 40s, on the cusp of Indian independence, Gopeshwar Pal developed a practice that obscured the division between 'artist' and 'artisan.' He succeeded in both revolutionizing Durga Puja images and creating highly regarded civic statuary. By the 1970s, practitioners from the other side of the divide known for their contributions to international modernism, including Paritosh Sen (1977), Meera Mukherjee (1979 and 1990), and Bikash Bhattacharjee (1991), added to the dynamism of Durga Puja by accepting commissions from the Bakul Bagan Puja association to create images of the goddess for their pandals, confirming that Durga Puja was a venue for 'art' as well as devotion, revelry, and commerce (Bhattacharya and Basu 2016) (Figure 13).

Nowadays, leading puja designers build art practices that transcend the divide between art, artisanry and commercial art, with work shown in art gallery spaces as well as puja pandals. Some have succeeded in moving Durga images from pandal to gallery-like exhibition spaces. Bhabatosh Sutar, a pioneer of this movement, created a Durga image in (fired) terracotta for the Barisha Shrishti Puja association in 2003 that was acquired for display at the five-star ITC Sonar Bangla hotel (Guha-Thakurta 2015, p. 265). The development of theme pujas and the implications for the categorization of art practitioners and their work were incisively explored in Tapati Guha-Thakurta's (2015) *In the Name of the Goddess* and the twenty-first century situation of mritshilpis and their practices is extensively presented in Moumita Sen's (2016) Ph.D. dissertation, *Clay modelling in West*

*Bengal: Between Religion, Art and Politics* and Geir Heierstad (2017), *Caste, Entrepreneurship and the Illusions of Tradition: Branding the Potters of Kolkata.*

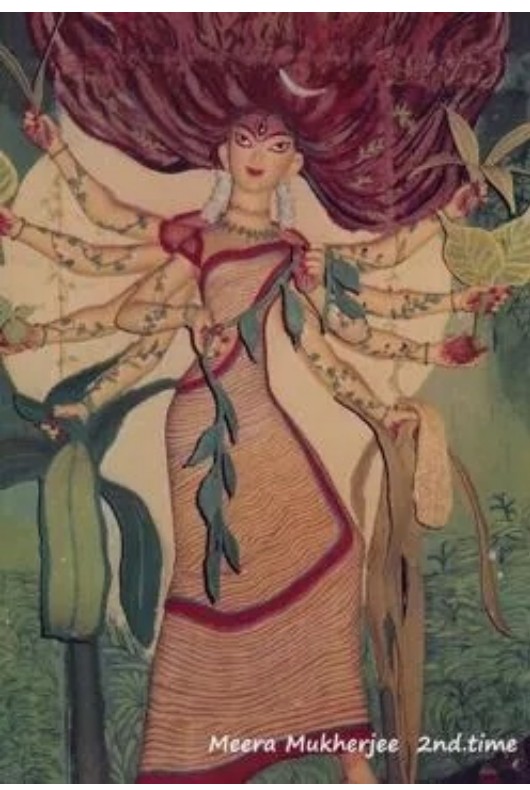

**Figure 13.** Meera Mukherjee (1923–1998), Durga for Bakul Bagan Sarbojanin Durga Puja, Kolkata, 1990. Image courtesy of Bakul Bagan Sarbojanin Durga Puja, published in *Hindustan Times* online accompanying article by Snigdhendu Bhattacharya and Arpit Basu. https://www.hindustantimes.com/kolkata/where-masters-of-art-bring-durga-to-life/story-gzTx6wGfjtWNYzhAb3dAyK.html (accessed on 1 May 2022).

The historical trajectory of terracruda sculpture in Bengal, sketched in this essay, demonstrates how clay has been a "player in the world" from the 1600s when the practice of terracruda sculpture emerged into prominence. Over centuries, clay sculptors and their clients have benefitted from clay's agile plasticity—its abilities to take on subtle and dramatic alterations of form. They took advantage of terracruda's penchant to disintegrate for developing a cycle of making, using, and disposing that supported an annual round of religious festivals in harmony with the progression of seasons and the phases of the moon. Devotees of the goddess, acknowledging her identity with clay-earth, recognized the potency of images that embodied her earthy nature. Terracruda's ephemerality ensured perennial opportunities for making that foregrounded iconographic form by requiring that representations of deities be continually reproduced. Clay's plasticity imparted flexibility to portrayals that could affirm an established form through replication or enable subtle and dramatic reinterpretation. Mritshilpis and their clients deployed this flexibility to influence the development of the devotional worship and to reinforce or shift political and social hierarchies. As the pace of change in all quarters accelerated with the onset of British rule, the appetite for innovation rose, positioning mritshilpis as interpreters, re-envisioning deities in response to new circumstances, and deploying terracruda sculpture in shong figures and decor to playfully engage with contemporary social life and to appropriate and indigenize foreign cultural icons, from characters of European literature to football stars from Brazil, Argentina, and Portugal (Indo-Asian News Service 2014). In this region, mundane and fragile terracruda sculpture became a significant participant, contributing to the reshaping of a rapidly changing cultural landscape (Figure 14).

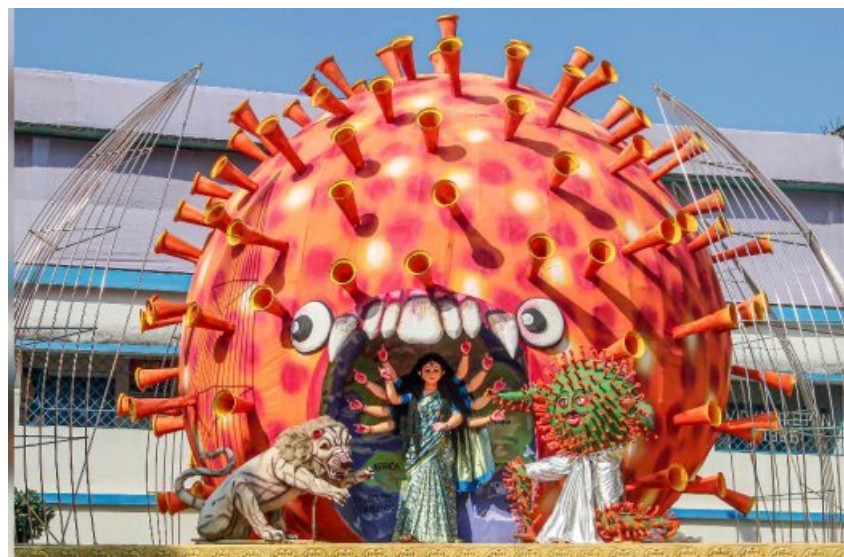

**Figure 14.** Coronavirus-themed public puja pandal at Balurghat in South Dinajpur district, West Bengal, October 18, 2020 Credit: PTI Photo. https://www.deccanherald.com/national/durga-puja-woes-rise-on-covid-19-front-as-daily-death-toll-crosses-1000-mark-again-903816.html (accessed on 1 May 2022).

**Funding:** This research was funded in part by the Asian Cultural Council in 2016.

**Data Availability Statement:** Not applicable.

**Conflicts of Interest:** The author declares no conflict of interest.

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
