# Peer review of "Making, Using, Disposing, Remaking…: Sacred Arts of Re-Creation in Southern Asia"

_religions, doi:10.3390/rel13070657_

Round 1

Reviewer 1 Report

The author discusses how the perennial re-creation of terracruda ritual images in the eastern Indian subcontinent supported the rise of goddess worship. Especially focusing on the images of the goddess Durga, the author tries to elucidate how these images stimulated the expansion of the annual cycle of religious festivals and contributed to a modernizing cosmopolitan public culture in the eastern region of India. It is a very intriguing subject, and new to me. However, for better arguments, I recommend the author revise a few parts of the manuscript.

As fundamental sculptural mediums, the author explained the plasticity and fluidity of clay used for the terracruda in the region(Line 112-142, 243-256, 292-300). Everyone is familiar with the properties of clay, and I don’t see any particular singularity in the earth of eastern India and Bangladesh. It is recommended to reduce repetitive explanations to a minimum.

In the case of Raja Krishnachandra, the author argues that the immersion of terracruda images after worship ensured a perennial demand for new images and stimulated makers to hone their skills and aesthetic finesse. Even if it could be a possible interpretation, It needs to be clarified with a more probable argument.

It would be better to explain in detail what the autumn puja means to the believers in the region, not to the terracruda artisans, other than doing it once more. (Line 314-317)

Overall this manuscript shows a well-organized composition and thorough research. However, considering that this manuscript has covered a fairly long period from the 1600s to the modern times, at the beginning of the paper, it would be good to suggest why this long time should be discussed.

Author Response

Thank you for your suggestions. 

I have revised the introduction to emphasize that the long period from 1600 to the present illustrates how the nimble plasticity of clay facilitated ongoing alterations of representations of the goddess and other deities, as well as non-sacred depictions of heroes, celebrities and scoundrels, that enabled figural forms to respond to the enormous shifts in regional circumstances from being a Mughal province, to a British colony, to a sovereign nation moving from state socialism towards an accommodation with capitalism.

I did not follow the interpretation of repetition regarding lines 112-142, about the intrinsic physical capabilities of terracruda, lines 243-256 which exemplify how terracruda's physical character facilitates creating new manifestations of the goddess and technical adaptations that accommodated a social spectrum of devotees from rulers to common folk, and lines 292 -300 which are concerned with Raja Krishnachandra's program to strengthen devotion to the goddess to garner her support for local rulers, to elicit their subjects' support for their rule, and to bolster their standing among the region's politically powerful.

I do not agree that "everyone is familiar with the properties of clay". Terracruda sculpture has long been overlooked, confused with terracotta and/or stucco. With the exception of K. M. Varma's (largely ignored) 1970 book, it is only in the past two decades that attention has been paid to the prevalence and significance of air-dried, unfired terracruda (for example Bean 2011, 2017, Luczanits 2004, Kaur 2005, Boivin 2008, Sen 2016, Heierstad 2017). It is also the case that local variations in clays greatly determine their capacities for firing and for modeling. I have tried to clarify some of these points in response. 

I have also tried to be clearer that Krishnachandra and other zamindars were motivated to create an important autumn Durga puja in order to garner the goddess’s support for righteous rulers as Rama had done in the Bengali Ramayana, and to promote their standing among the politically powerful in the region with their lavish annual Pujas, as well as to build support among their subjects and encourage them to worship the goddess. The point is that ephemeral terracruda ritual images served these purposes well in part because of the necessity of recreating them for each occasion. 

I am grateful for your input and hopeful that the revisions will clarify the argument offered. 

Reviewer 2 Report

Overall this is a good article, but it requires some revision. In particular:

Not all of the references in the text are listed in the references list at the end (for example Sastri 1960, on line 174 and Nicholas 2013 on line 181).

There is inconsistency in use of italics - terms are introduced in italics, but subsequently not italicised (so zamindars on line 32, but zamindars subsequently, and mritshilpis on line 230 but mritshilpis subsequently)

There are other minor typograhical erros, such as the punctuation of lines 38 and 39.

There is inconsistency - reference to Kolkata in lines 377 and 378 but Calcutta in line 154 and elsewhere

The point about the plasticity of clay could be further emphasised, not least in relation to the shift to other materials (fibreglass etc) that is referenced towards the end of the article.

Author Response

Thank you for pointing out the missing references, Sastri 1960 and Nicholas 2013. I have added these and rechecked the full manuscript.

I used italics for non-English words, such as zamindar and mritshilpi, both of which have no satisfactory English translation, according the the Chicago Manual of style: 

11.3: Non-English words and phrases in an English context: Italics are used for isolated words and phrases from another language, especially if they are not listed in a standard English-language dictionary like Merriam-Webster’s Collegiate (see 7.1) or are likely to be unfamiliar to readers (see also 7.54).  If such a word or phrase becomes familiar through repeated use throughout a work, it need be italicized only on its first occurrence. If it appears only rarely, however, italics may be retained.

Thank you for noting the punctuation problems and prompting further checking.

About switching from Calcutta to Kolkata at the end of the paper in the section on recent developments. I noted that the city's official name became Kolkata in 2001.

I tried to respond to the comment about emphasizing plasticity.